# Rice Breeding in Russia Using Genetic Markers

**DOI:** 10.3390/plants9111580

**Published:** 2020-11-15

**Authors:** Elena Dubina, Pavel Kostylev, Margarita Ruban, Sergey Lesnyak, Elena Krasnova, Kirill Azarin

**Affiliations:** 1Federal Scientific Rice Centre, Belozerny, 3, 350921 Krasnodar, Russia; lenakrug1@rambler.ru (E.D.); arrri_kub@mail.ru (M.R.); Sergei_l.a@mail.ru (S.L.); 2Agrarian Research Center “Donskoy”, Nauchny Gorodok, 3, 347740 Zernograd, Russia; krasnovaelena67@mail.ru; 3Department of Genetics, Southern Federal University, 344006 Rostov-on-Don, Russia; azarinkv@sfedu.ru

**Keywords:** rice, salinity, submergence tolerance, blast, SSR markers, PCR analysis

## Abstract

The article concentrates on studying tolerance to soil salinization, water flooding, and blast in Russian and Asian rice varieties, as well as hybrids of the second and third generations from their crossing in order to obtain sustainable paddy crops based on domestic varieties using DNA markers. Samples IR 52713-2B-8-2B-1-2, IR 74099-3R-3-3, and NSIC Rc 106 were used as donors of the *SalTol* tolerance gene. Varieties with the *Sub1A* locus were used as donors of the flood resistance gene: Br-11, CR-1009, Inbara-3, TDK-1, and Khan Dan. The lines C101-A-51 (Pi-2), C101-Lac (Pi-1, Pi-33), IR-58 (Pi-ta), and Moroberekan (Pi-b) were used to transfer blast resistance genes. Hybridization of the stress-sensitive domestic varieties Novator, Flagman, Virazh, and Boyarin with donor lines of the genes of interest was carried out. As a result of the studies carried out using molecular marking based on PCR in combination with traditional breeding, early-maturing rice lines with genes for resistance to salinity (*SalTol*) and flooding (*Sub1A*), suitable for cultivation in southern Russia, were obtained. Introgression and pyramiding of the blast resistance genes *Pi-1, Pi-2, Pi-33, Pi-ta*, and *Pi-b* into the genotypes of domestic rice varieties were carried out. DNA marker analysis revealed disease-resistant rice samples carrying 5 target genes in a homozygous state. The created rice varieties that carry the genes for blast resistance (Pentagen, Magnate, Pirouette, Argamac, Kapitan, and Lenaris) were submitted for state variety testing. The introduction of such varieties into production will allow us to avoid epiphytotic development of the disease, preserving the biological productivity of rice and obtaining environmentally friendly agricultural products.

## 1. Introduction

Rice (*Oryza sativa* L.) is the most important food crop for more than half of the world’s population (China, Japan, India, Bangladesh, etc.). Biotic and abiotic stressors are the main obstacles to increasing global crop production and expanding rice production. It was found that only about 10% of the world’s agricultural land is located in areas that do not suffer from stress factors [1].

Decreased rice yields in adverse climatic conditions threaten global food security. Genetic loci that ensure productivity in difficult conditions exist in the germplasm of cultivated plants, their wild relatives, and species adapted to extreme conditions [2].

One-fifth of the world’s irrigated land (North Africa, Central and South-East Asia, etc.) is adversely affected by high soil salinity [3]. About 45 million hectares in the world are subject to soil salinization [4]. In the Russian Federation, rice is grown on an area of about 200 thousand hectares, of which about 80 thousand hectares are saline [5]. The decline in productivity on saline soils can be overcome by developing rice varieties tolerant to salinity and introducing them into agricultural production. Several non-allelic genes provide tolerance to salinity during ontogenesis [6]. The main locus of salt tolerance is *SalTol*, which was first identified in some rice varieties [7,8]. This locus is mapped on chromosome 1 and its main function is to control the balance of Na+/K+ ions in rice plants [9].

One of the serious abiotic stress factors for rice, which inhibits plant growth and affects crop yield, is prolonged submersion of plants under water, which often happens to large areas of land in the rice-growing regions of South-East Asia [5]. Rice dies if total flooding lasts more than two weeks. A negative effect on the growth and development of rice plants at this time is exerted by a lack of oxygen (O_2_) and limited diffusion of carbon dioxide (CO_2_). Lack of light due to turbid flood water in the rice paddies during this period limits the ability of plants to photosynthesize and can even lead to their death [4,5,6].

Scientists in Asia have found the *Sub1A* gene, which regulates the response of plant cells to ethylene and gibberellin, leading to restriction of carbohydrate intake and dormancy of shoots under water, which contributes to tolerance to immersion [10,11]. In Russia, this gene can be used to develop varieties resistant to a large layer of water during the germination phase, which will become an effective way to protect rice from weeds without herbicides. Most weeds die under water without oxygen, and rice can survive. To develop such varieties, it is necessary to combine in one genotype genes with increased energy of initial growth, the ability to anaerobic germination, resistance to prolonged flooding and lodging.

In all countries of the world, including Russia, blast is among the most dangerous fungal diseases of rice and causes large yield losses in the years of epiphytoty. The most effective way to protect rice without fungicides is to grow blast-resistant varieties. More than 50 genes of resistance to this pathogen are known: *Pi-1, Pi-2, Pi-33, Pi-b, Pi-ta, Pi-z*, etc. [12]. Combining several effective resistance genes with their contribution on the genetic basis of the best varieties is an effective breeding strategy for resistance to variable fungal pathogens. Lines with a combination of 3–5 resistance genes show an increase and broadening of the spectrum of blast resistance in comparison with lines with separate genes. A number of successful breeding programs have already been carried out abroad to develop blast-resistant rice varieties by the gene pyramiding method using marker breeding [13].

Resistance to various biotic and abiotic factors is one of the traits that are difficult to assess when the assessment of the breeding material is possible only in the presence of an appropriate stress factor. At present, during the breeding of agricultural plants for resistance, the splitting population obtained from the crossing of resistance sources with genotypes that have productivity is tested against a natural background, or artificial infection is carried out under controlled conditions. This procedure, although it gives excellent results, is quite lengthy and costly. In addition, there are always susceptible plants that have escaped damage [14].

The use of DNA markers allows us to speed up the assessment and conduct selection without phenotypic assessment, at an early stage, regardless of the external conditions. In recent years, great progress has been made in the development of molecular marking technologies and their application to control complex agronomic traits using marker breeding [15]. The technology of molecular marking of resistance loci makes it possible to quickly select plant forms with target genes without using provocative backgrounds [16]. The identification of molecular markers linked to genes of resistance to these factors facilitates breeding work. The use of DNA markers brings the breeding of agricultural plants to a qualitatively new level, making it possible to evaluate genotypes directly and not through phenotypic manifestations, which, ultimately, is realized in the accelerated development of varieties with a complex of valuable traits [14]. Therefore, it is relevant to develop new rice varieties by marking [17].

The purpose of the study was the development of initial rice material using DNA markers for breeding highly productive varieties resistant to biotic and abiotic environmental stress factors: soil salinity, prolonged flooding, and blast.

## 2. Materials and Methods

We used samples from the collection of the Institute of Agricultural Genetics (Vietnam) as donors of the transferred salt tolerance gene: IR 52713-2B-8-2B-1-2, IR 74099-3R-3-3, and NSIC Rc 106, which were crossed with the early--maturing Krasnodar variety Novator. These varieties carry the *SalTol* locus, which has been mapped near the centromeric region of the first chromosome. RM493 and RM7075 [18] were used as flanking SSR-markers of this locus, with the greatest difference in the length of microsatellite repeats between the parental forms.

Varieties with the *Sub1A* locus were used as donors of the flooding resistance gene: BR-11, CR-1009, Inbara-3, TDK-1, and Khan Dan. The early-ripening variety Novator and rice lines with the introgressed genes for blast resistance *Pi-2* and *Pi-33* were also taken as recipients. The *Sub1A* locus is mapped to an interval of 0.06 morganides in chromosome 9 [11]. We used microsatellite markers for the *Sub1A* gene, CR25K and SSR1A. The *Sub1A* gene was identified by molecular marking based on PCR using specific primers.

When transferring blast resistance genes, lines C101-A-51 (*Pi-2*), C101-Lac (*Pi-1, Pi-33*), IR-58 (*Pi-ta*), and Moroberekan (*Pi-b*) were used. To identify the Pi-1 gene, we used primer pairs of the flanking microsatellite SSR markers RM224 and RM144; for the Pi-2 gene, we used RM527 and SSR140; for the Pi-33 gene, RM310 and RM72; for the *Pi-b* and *Pi-ta* genes, intragenic markers developed in the laboratory of biotechnology, Federal Scientific Rice Centre. They are localized on chromosomes 11, 6, 8, 2, and 12, respectively (Table 1) [19,20].

The early-ripening released rice varieties Boyarin, Flagman, and Virage served as the paternal form. During plant hybridization, pneumocastration of maternal forms and pollination by the Twell method were used [21]. Hybrid plants were grown on checks of Federal State Unitary Enterprise “Proletarskoe” (Rostov region) and the Federal State Unitary Elite Seed-growing Enterprise “Krasnoe” of the Federal Scientific Rice Centre, Krasnodar region. From the selected rice leaves, genomic DNA was isolated under laboratory conditions at the Federal Scientific Rice Centre, the Academy of Biology and Biotechnology of the Southern Federal University, and the All-Russian Research Institute of Agricultural Biotechnology. PCR products were separated by electrophoresis in 2.5% agarose and 8% acrylamide gels. The experimental data were statistically processed using Microsoft Excel and Statistica 6 software.

The account of the degree of damage to plants (in percentages) was carried out on the 14th day after inoculation, in accordance with the express method for assessing rice varietal resistance to blast. The assessment was carried out by taking two indicators into account: the type of reaction (in points) and the degree of damage (in percentages), using the ten-point scale of the International Rice Research Institute [12]:resistant: 0–1 point—no damage, small brown spots, covering less than 25% of the total leaf surface;medium resistant: 2–5 points—typical elliptical blast spots, 1–2 cm long, covering 26–50% of the total leaf surface;susceptible: 6–10 points—typical blast spots of elliptical shape, 1–2 cm long, covering 51% or more of the total leaf surface.

The intensity of disease development (IDD, %) was calculated by the formula (Equation (1)):IDD = ∑ (a × b)/n × 9(1)
where IDD is the intensity of disease development (%), ∑ (a × b) is the sum of the products of the number of infected plants multiplied by the corresponding damage point, and n is the number of recorded plants (pcs).

Depending on the damage points, all varieties wee conventionally divided into 4 groups: resistant, intermediate, susceptible, and strongly susceptible.

## 3. Results and Discussion

The development of blast-resistant varieties and their rapid introduction into production is the most promising solution in the fight against this disease. However, the development of resistant varieties is one of the most difficult areas of breeding. The causative agents of the disease have a great potential for variability, which, combined with its colossal reproduction capabilities, provides the pathogen with the highest adaptive capabilities. Combining several effective genes of resistance on a genetic basis of the best varieties widely used in production is an effective breeding strategy for long-term resistance to variable fungal pathogens.

Based on the use of DNA marker breeding (marker-assisted selection (MAS)—breeding with use of DNA markers towards genes of interest), we introduced 5 blast resistance genes into domestic rice varieties adapted to the agro-climatic conditions of rice cultivation in southern Russia.

A series of crosses made it possible to obtain rice lines based on the varieties Boyarin, Flagman and Virage with the introgressed and pyramided blast resistance genes *Pi-1, Pi-2, Pi-33, Pi-ta*, and *Pi-b* in a homozygous state. During all cycles of backcrossing, the transfer of the dominant alleles of each such gene in the offspring was controlled by closely linked molecular markers. Plants with no resistance alleles in the genotype were discarded.

At the first stage of work in 2005 at Agrarian research center “Donskoy”, 6 hybrids were obtained from crossing the varieties Boyarin and Virage with three donors of blast resistance carrying the *Pi-l, Pi-2*, and *Pi-33* genes. After analysis at the Federal Scientific Rice Centre, homozygous forms were identified for the dominant alleles.

At the second stage of work (2008), after crossing the *Pi-1 + 33* × Boyarin and *Pi-2* × Boyarin hybrids between themselves, it was possible to obtain forms with three pyramided genes simultaneously: *Pi-1, Pi-2*, and *Pi-33* in a homozygous state.

At the third stage of work (2010), they were hybridized with varieties—donors of the *Pi-ta* and *Pi-b* genes—for combining 5 genes. There were two types of crosses: ((*Pi-1 + 2 + 33*) × *Pi-ta*) × *Pi-b* and *Pi-1 + 2 + 33* × (*Pi-ta* × *Pi-b*).

Leaves were selected from the best F_2_ hybrid plants for DNA analysis at All-Russian Research Institute of Agricultural Biotechnology and the Federal Scientific Rice Centre using one marker for each gene. Based on the analysis results, it was possible to isolate two rice samples that were homozygous for all five dominant alleles. Reanalysis of the leaves of these samples confirmed last year’s results, i.e., homozygosity for the dominant alleles of all five loci.

Figure 1 and Figure 2 show the panicles of two lines, 1225/13 and 1396/13, which show the presence of dominant alleles at five loci in the homozygous state: *Pi-1*, *Pi-2*, *Pi-33*, *Pi-b*, and *Pi-ta*.

Line 1225/13 is early maturing, matures in 110 days, and dwarfish (80 cm), with a small panicle (15 cm) (Figure 1).

The second line 1396/13 is mid-ripening, the period to maturity is 120 days, and it is taller (100 cm), with a large long panicle (22 cm) (Figure 2).

Against the infectious background in the Federal Scientific Rice Centre, the index of disease development (IDD) in this line was only 1.4%, while the variety Novator was damaged by 67.7%. The results of the analysis made it possible to send these lines to the breeding nursery in 2014–2015 for testing for yield and blast resistance. The variety **Pentagen** (1396/13), carrying 5 genes for blast resistance, is widely used in hybridization with high-yielding Russian varieties.

In the process of work at the Federal Scientific Rice Centre in 2007–2008, crosses were carried out and F_1_ hybrids were obtained from the combination (Flagman × C101-Lac) × (Flagman × C-101-A-51), which have the blast resistance genes *Pi-33* and *Pi-2* in their genotypes, respectively. The resulting F_1_ generation was used in backcrosses with the recipient parental forms. It should be noted that the F_1_ plants had a high degree of sterility (up to 95%). After the first series of backcrosses in 2008, the ΒC_1_ and ΒC_2_ generations were obtained in artificial climate chambers. In BC_1_ populations, fertility increased and averaged about 50%. Starting from the first backcrossing, marker control was carried out for the presence of transferred donor alleles in the hybrid offspring. In 2009, plants of the ΒC_3_ and BC_4_ generation were obtained. Among these plants, we selected the forms with the shortest growing season and the highest panicle fertility. From the ΒC_4_F_1_ stage (the first self-pollination of rice plants, which makes it possible to transfer the donor allele to a homozygous state), individual selection was carried out. Segregation for Pi-2, Pi-33, and Sub1A genes fit into the Mendelian framework: in the second generation as a result of DNA analysis of the obtained plants, the ratio was 1:2:1 by genotype and 3:1 by phenotype.

Plants were selected that were closest in morphotype to the recipient parental form and had donor genes for resistance to the pathogen *Pyricularia oryzae* Cav. in their genotype in a homozygous state [22].

Figure 3 shows the results of PCR analysis for identification of the *Pi-33* blast resistance gene in the ΒC_4_F_3_ hybrid material.

The figure shows that plants Number 2, 4, and 7–12 are homozygotes for the dominant allele; plants Number 1 and 3 are heterozygous. The size of the PCR product in varieties with the dominant allele of the Pi-33 gene, which determines the resistance, is 198 bp; in varieties with a recessive allele, it is 152 bp.

In 2015–2016, the resulting rice lines with introgressed blast resistance genes *Pi-2* and *Pi-33* were crossed with the variety Khan Dan (Vietnamese breeding): the donor of the *Sub1A* gene. This work was performed for obtaining breeding material with combined genes for disease resistance and tolerance to prolonged immersion of plants under water. In 2017–2020, F_4_ and BC_2_F_3_ generations were obtained using climate chambers at the Federal Scientific Rice Centre (All-Russian Rice Research Institute, Krasnodar, Russia).

To increase economic efficiency and reduce labor costs, multi-primer systems have been developed to identify two genes (*Pi* and *Sub1A*) in a hybrid material simultaneously.

At the first stage, when we selected DNA markers for reliable interpretation of PCR products and identification of non-specifically amplified fragments, the following parameters were taken into account: the annealing temperature of specific pairs introduced into the reaction mixture, the difference in the size of PCR products synthesized during amplification with specific primer pairs (at least 100 base pairs), and the self-complementarity of the primer sequences.

The results of testing the combination of primer pairs flanking the marker regions of the *Pi-2 + Sub1A* genes are shown in Figure 4.

The electrophoregram shows that when PCR with such a combination of molecular markers is carried out, the target products specific for DNA markers of the desired genes are reliably amplified. Samples Number 3 and 12 have dominant alleles of the genes *Pi-2* and *Sub1A* in a homozygous state in their genotype; Samples 1, 4, 5, and 9 are homozygous for the *Sub1A* gene and have the *Pi-2* gene in the genotype in a heterozygous state; Sample 10 is a recessive homozygote for two target genes and was rejected. The size of the PCR product in varieties with the dominant allele of the *Pi-2* gene, which determines the resistance, is 233 bp. The size of the PCR product in varieties with the dominant allele of the *Sub1A* gene, which determines the resistance, is 118 bp. Clear identification on the electrophoregram makes it possible to reliably identify the presence of dominant alleles of the target genes.

The introduction of such varieties into production will allow us to avoid epiphytic development of the disease, preserving the biological rice productivity, and obtaining environmentally friendly agricultural products.

**Magnat** is the first cultivar in Russia created at the Agrarian Research Center Donskoy together with the Federal Scientific Rice Centre by the method of marker selection from a hybrid population (C101A-51 × Boyarin) × (C101 LAC × Boyarin) with transfer of blast resistance genes. Sample C101 LAC is a donor of the genes *Pi-1* and *Pi-33*, and C101A-51 is a *Pi-2*. The growing season is 125 ± 1 days and the plant height is 96 ± 2 cm. The panicle is erect and compact, 17.5 ± 0.5 cm long, and bears 185 ± 5 spikelets. The grain is oval, 8.3 ± 0.2 mm long, 3.1 ± 0.1 mm wide, and 2.2 ± 0.1 mm thick and weighs 24.0 ± 2.0 mg. The yield of the Magnat variety was 8.25 t/ha, which is 1.1 t/ha higher than that of the Boyarin standard.

The rice variety **Pirouette** was bred at the Agrarian Research Center Donskoy, together with the Federal Scientific Rice Centre, by the method of stepwise hybridization and marker breeding from a hybrid population (C101-A-51 (*Pi-2*) × Boyarin) × (C101-Lac (*Pi-1 + 33*) × Virazh). It contains three blast resistance genes: *Pi-1, Pi-2,* and *Pi-33.* The variety is mid-ripening, the growing season from flooding to full ripeness is 124 ± 1 days. The average yield of the variety Pirouette was 9.57 t/ha, which is 1.13 t/ha higher than that of the standard variety Yuzhanin. Plant height is 88 ± 2 cm; the panicle is erect, compact, and 17.5 ± 0.5 cm long and carries 165 ± 5 spikelets. The spikelets are oval, 8.9 ± 0.2 mm long, and 3.7 ± 0.1 mm wide. The weight of 1000 grains is 31.6 ± 2.0 g. The variety is resistant to lodging and shedding, is cold-tolerant, and germinates well from under a layer of water. It has been ncluded in the Register of Breeding Achievements of the Russian Federation for the North Caucasus region since 2020.

The rice variety **Kapitan** was bred at the Agrarian Research Center Donskoy in cooperation with the Federal Scientific Rice Centre by the method of triple backcrossing and marker breeding from the Flagman × IR-36 hybrid population. The variety is mid-ripening and the growing season from the flooding to full ripeness is 120 ± 1 days. On average, over the years of competitive testing, the yield of the variety Kapitan was 8.13 t/ha, which is 0.64 t/ha higher than that of the variety Yuzhanin. A higher yield of this variety is formed due to more grain in the panicle and an increased weight of the caryopsis. The average height of plants is 90 ± 2 cm; the panicle is semi-inclined, compact, and 18.5 ± 0.5 cm long; and the average number of spikelets is 140 ± 10 pieces (Figure 5). The grains are oval, 9.5 ± 0.2 mm long, and 3.6 ± 0.1 mm wide. The average weight of 1000 grains is 35.0 ± 2.1 g. The variety carries the *Pi-ta* gene and is resistant to blast, lodging, and shedding. The variety has been under state testing since 2019.

The rice variety **Argamak** was bred at the Agrarian Research Center Donskoy by individual multiple selection of plants with the largest panicles from a hybrid population Il. 14 (*Pi-1, Pi-2, Pi-33*) × Kuboyar. The variety belongs to the mid-ripening group, and the growing season from flooding to full ripeness is 119 days. On average, over the years of competitive testing (2017–2019), the yield of the variety was 8.79 t/ha, which is 1.59 t/ha higher than that of the variety Yuzhanin. The maximum yield was formed in 2019: 10.1 t/ha, 2.55 more than the standard. The high yield of this variety was formed due to the greater grain content of the panicle than that of the standard and the increased density of the stem. Plant height is 93 ± 2 cm on average; the panicle is erect, compact, and 16 ± 0.5 cm long; the average number of spikelets is 142 ± 6 pieces. The grains are oval, 8.4 ± 0.2 mm long, 3.3 ± 0.1 mm wide. Weight of 1000 grains—31.1 ± 1.9 g. The variety is resistant to blast, lodging, and shedding. It has been tested at state varietal testing since 2020.

The rice variety **Lenaris** (Federal Scientific Rice Centre) had shown high adaptability, non-lodging, and the possibility for straight combine harvesting. Its yield was 10.6 t/ha. Plants had high spikelet fertility and short stems (77 ± 5 cm) and were resistant to the Krasnodar population of *P. oryzae* as well. Their panicle is slightly drooping and compact; its length is 18 ± 1.0 cm. The mass of 1000 grains is about 30.4 ± 1.8 g.

In 2013–2014, the Agrarian Research Center Donskoy conducted crosses and obtained F_1_–F_2_ hybrids of the variety Novator with Asian donor rice varieties carrying the *SalTol* and *Sub1A* genes. The hybrids of the second generation varied significantly in terms of quantitative traits: growing season (from early ripening to non-flowering), plant height (75–122 cm), panicle length (14–25 cm), number of filled grains (80–206 pcs), number of spikelets (99–300 pcs), panicle density (4.4–16.6 pcs/cm), 1000-grain weight (26.3–34.9 g), grain weight per panicle (2.1–5.5 g), etc.

Hybridization of the salt-sensitive domestic variety Novator with the lines IR52713-2B-8-2B-1-2, IR74099-3R-3-3, and IR61920-3B-22-2-1 (NSIC Rc 106)—*SalTol* locus donors—was carried out. The first generation of hybrids was used to generate an F_2_ hybrid population. From the populations of plants of the second generation, 90 early-ripening samples with well-ripened grains (30 in each combination of crossing) were selected, which were analyzed by PCR for the presence of introduced *SalTol* alleles. As an example, Figure 2 shows the data of electrophoretic analysis of PCR products with the Rm493 marker. The donor allele of the parental line NSIC Rc 106, designated as 2.2, was found in a homozygous state in Sample 282. The rest of the plants, whose amplification spectra are presented in this electrophoregram, carried the alleles of the donor and the variety Novator; that is, they were heterozygous for the *SalTol* locus (Figure 2). Similar results were obtained during DNA analysis of the studied rice samples with the RM7075 marker (Figure 6 and Figure 7).

In general, according to the results of DNA analysis of F_2_ hybrids, 17 plants homozygous for the dominant allele of the *SalTol* locus were identified, 29 samples carried *SalTol* in a heterozygous state, and 44 plants showed only recessive alleles inherited from the variety Novator.

Segregation for *SalTol* genes did not fit into the Mendelian framework, since the sample was unrepresentative due to selection. Plants with recessive alleles of the gene prevailed, and the number of salt-tolerant dominant homozygotes was less than the expected number. This is due to the linkage of *SalTol* genes with genes unfavorable for plants in our conditions: photosensitivity, late maturity, spikelet shedding, and spinosity.

Testing plants under salinity in the early stages of development is a quick, common method based on simple criteria. It was shown that at the initial vegetation stage, the length of the root and shoots and seed germination are potential indicators of resistance to the effects of increased salt concentrations [18,19]. Evaluation of the potential salt tolerance of the studied rice hybrids and their parental forms revealed significant variations in salinity tolerance depending on the genotype. The greatest decrease in seed germination—52%—was found in the salt-sensitive variety Novator. The line NSIC Rc 106 and second-generation plants, which were homo- and heterozygous for the *SalTol* locus, showed the highest resistance by seed germination (germination decrease of less than 5%). The donor lines IR52713-2B-8-2B-1-2, and IR74099-3R-3-3 and hybrid combinations obtained on their basis with the *SalTol* gene in a homozygous state also showed high resistance for this trait.

The least suppression of growth indices, as well as in the case of seed germination, was recorded in the lines NSIC Rc 106, IR52713-2B-8-2B-1-2, IR74099-3R-3-3, and *SalTol* homozygous plants from the F_2_ generation; the greatest decrease in the length of shoots and roots under salt stress was shown in the variety Novator and in hybrid plants that did not inherit the *SalTol* locus according to molecular analysis data. Thus, DNA analysis made it possible to simplify the breeding scheme and obtain salt-tolerant F_2_ hybrids carrying the *SalTol* locus in a homozygous state. These results indicate that the developed codominant markers of the *SalTol* locus RM 493 and RM 7075 are an effective tool for marker-assisted selection of salt-tolerant forms based on domestic rice genotypes.

Rice samples with *SalTol* genes in 2018–2020 were studied in a control nursery and in competitive variety testing; productive forms were identified.

At the same time, in 2013, hybrids were obtained by crossing the variety Novator with donors of the *Sub1A* gene. The Asian varieties turned out to be late-ripening and photosensitive and did not flower under our conditions. Hybridization was carried out only with the help of artificial climate chambers. The first generation in 2013 was characterized by a high degree of sterility (90–95%) and brown color of the flowering scales during maturation, which indicates significant genetic differences between the parental forms. In the second generation in 2014, a very large spectrum of splitting was observed in terms of the growing season, plant height, panicle length and shape, number of spikelets, and spinosity (Table 2).

This wide range of variability is not observed in other crops. This is due to the genetic and ecological-geographical remoteness of the crossed forms. In each combination, about 400 plants were selected for morphometric and genetic analysis. Among the F_2_ hybrids, we managed to select the best plants according to many traits, combining early maturity, optimal plant height, good grain size in panicles, non-shattering, and fertility of spikelets (Table 3).

PCR analysis of leaves was carried out in 20 plants of each of the four hybrids, as a result of which, forms with the *Sub1A* flood resistance gene were isolated. The electrophoretic analysis of PCR products with the RM 7481 marker is shown in Figure 8. The donor allele of the parental line CR-1009 was homozygous in Samples 2, 3, 5, 9, 13 and 17. Plants 2, 4, 6–8, 10, 11, 16, 18, and 19 were heterozygous at the *Sub1A* locus; that is, they carried both the alleles of the donor and the recessive alleles inherited from the variety Novator. Thus, according to the results of PCR analysis with the RM 7481 marker, 14 homozygotes of F_2_ plants at the *Sub1A* locus were identified, 40 samples carried *Sub1A* in a heterozygous state, and 22 plants inherited only the recessive allele from the variety Novator.

Of the analyzed BR-11 × Novator hybrid plants, the Sub1A gene (in homo- and heterozygous state) was present in nine, i.e., in a ratio of 9:11, although with monohybrid segregation, it should have been 15:5. In the hybrid combination CR-1009 × Novator, F_2_ segregated in a ratio of 18:2, i.e., almost all of the selected plants had the *Sub1A* gene. In the hybrids Inbara-3 × Novator and TDK-1 × Novator, segregation took place in a ratio of 14:6 or approximately 3:1, i.e., close to Mendelian.

The deviations in segregation of the two combinations can be explained by the influence of selection and gene linkage. A total of 55 plants with the target gene in the homo- and heterozygous state were isolated from 80 plants of four hybrids. The selected samples with the *Sub1A* gene in 2015 were reproduced in the Federal State Unitary Enterprise “Proletarskoye”of the Rostov Region, where the best F_3_ plants were selected from them for DNA analysis.

In F_3_ plants, significant morphological and biological segregation continued. Significant variation was noted for the growing season, plant height, size of panicles and caryopses, fertility, grain shedding, etc. The best forms were selected from them and leaves were taken for DNA analysis. At the next stage of work, in 2016–2020, constant lines carrying the *Sub1A* gene in a homozygous state were selected and tested for yield and resistance to prolonged water flooding. As a result, rice varieties for herbicide-free technologies will be developed, vigorously overcoming a deep layer of water in the germination phase with minimal seed loss.

## 4. Conclusions

1. As a result of the studies carried out using molecular marking based on PCR in combination with traditional breeding, early-maturing rice lines with genes for resistance to salinization (*SalTol*) and to flooding (*Sub1A*), which are suitable for cultivation in the south of Russia, were isolated.

2. Rice lines have been developed, the genotype of which contains five effective blast resistance genes (*Pi-1, Pi-2, Pi-33, Pi-ta,* and *Pi-b*). The introduction of such varieties into production will allow us to avoid epiphytotic development of the disease, preserving the biological productivity of rice and obtaining environmentally friendly agricultural products.

3. Samples of the F_4_ and BC_2_F_3_ generations were obtained with combined blast resistance (*Pi*) and prolonged flooding tolerance (*Sub1A*) genes as a factor in the control of weeds in the homo- and heterozygous state, which was confirmed by the data of their DNA analysis. The testing of the obtained rice breeding resources for submergence tolerance under laboratory conditions made it possible to select tolerant rice forms that will be studied in the breeding process for a complex of agronomically valuable traits. Their use will reduce the use of chemical plant protection products against diseases and weeds, thereby increasing the ecological status of the rice-growing industry.

The research was carried out with the financial support of the Kuban Science Foundation in the framework of the scientific project № 20.1/1.

## Figures and Tables

**Figure 1 plants-09-01580-f001:**
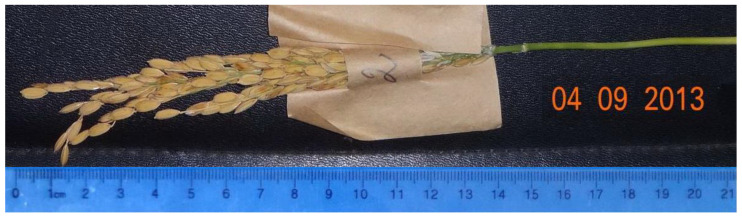
Panicle of the early-ripening line 1225/13.

**Figure 2 plants-09-01580-f002:**
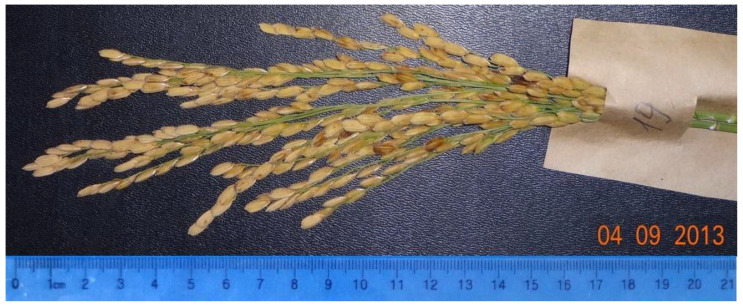
Panicle of the mid-ripening line 1396/13.

**Figure 3 plants-09-01580-f003:**
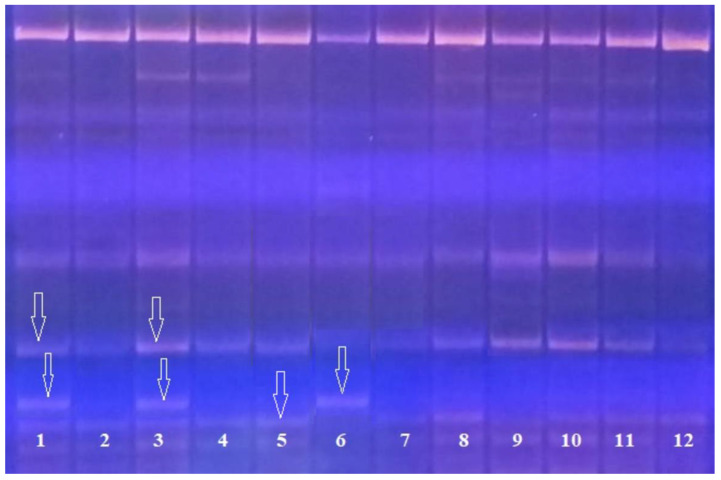
Electrophoregram of genomic DNA amplification products at the loci RM310 and RM72: 1–4, 7–12, analyzed hybrid ΒC_4_F_3_ plants; 5, donor line of the Pi-33 gene C101-Lac; Flagman, maternal form.

**Figure 4 plants-09-01580-f004:**
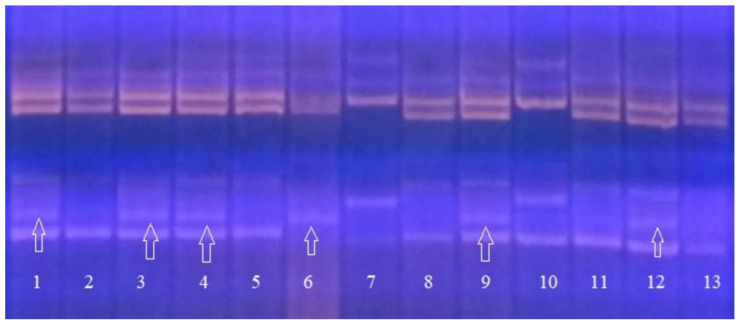
Electrophoregram of multiplex PCR of genomic DNA amplification products at the loci RM527 and SSR140 for the Pi-2 gene and at the Sub1A203 locus for the *Sub1A* gene: 1–5, 9–13, analyzed hybrid plants of the ΒC_2_F_3_ generation; 6, Khan Dan, donor of the *Sub1A* gene; 7, Flagman, maternal form; 8, C101Lac-A-51, donor line of the *Pi-2* gene.

**Figure 5 plants-09-01580-f005:**
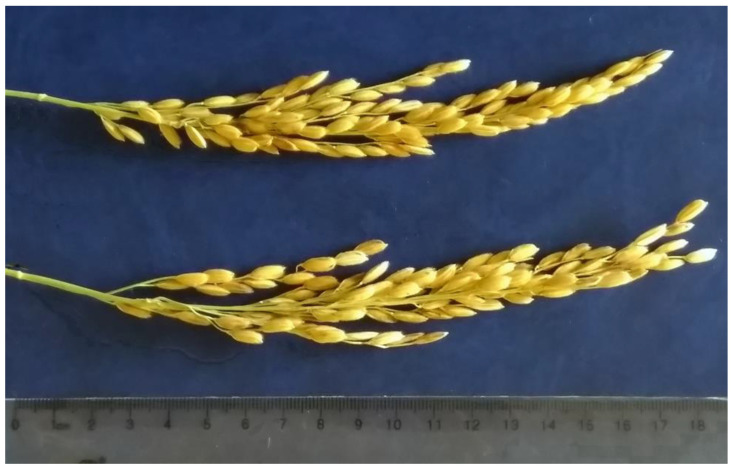
Rice panicles of the variety Kapitan.

**Figure 6 plants-09-01580-f006:**
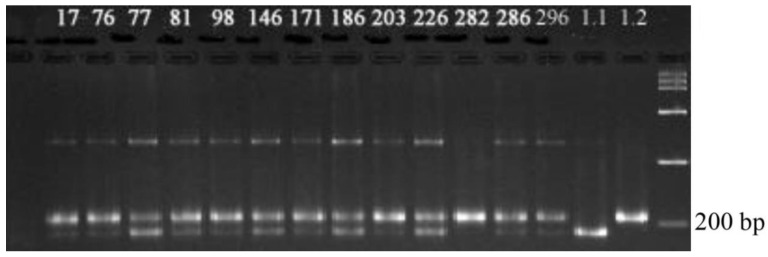
Electrophoregram of genomic DNA amplification products with RM 493: 1.1, Novator; 1.2, NSIC Rc 106; 17–296, hybrid plants NSIC Rc 106 × Novator; DNA marker (100–1500 bp).

**Figure 7 plants-09-01580-f007:**
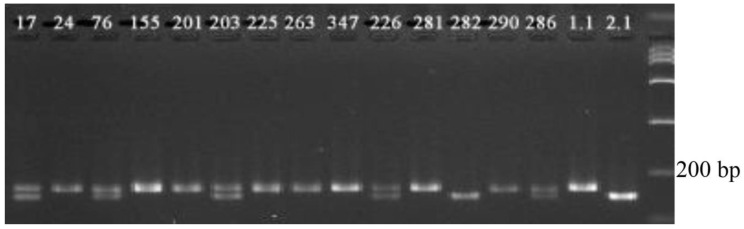
Electrophoregram of genomic DNA amplification products with RM 7075: 1.1, Novator; 2.1, NSIC Rc 106; 17–286, hybrid plants NSIC Rc 106 × Novator; DNA marker (100–1500 bp).

**Figure 8 plants-09-01580-f008:**
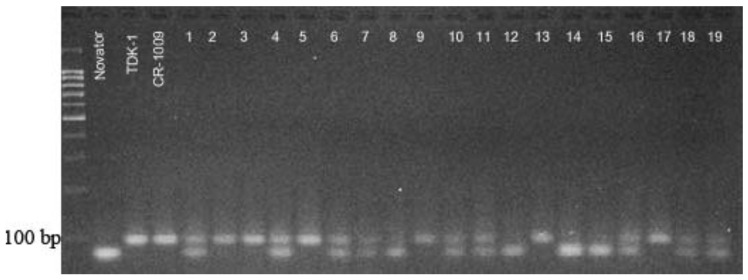
Electrophoregram of the amplification products of rice genomic DNA with the primer RM 4781. 1–19, F_2_ (Novator × CR-1009); TDK-1 and CR-1009, donor of the *Sub1A* gene. Molecular weight marker, 1 kb.

**Table 1 plants-09-01580-t001:** Nucleotide sequences of codominant markers for identification of the allelic status of genes Pi-1, Pi-2, Pi-a, and Pi-b.

Resistance Gene	Chromosomal Localization of Gene	Marker	Sequence	
*Pi-2*	6	Rm 527	F	GGC TCG ATC TAG AAA ATC CG
			R	TTG CAC AGG TTG CGA TAG AG
		SSR140	F	AAG GTG TGA AAC AAG CTA GCA
			R	TTC TAG GGG AGG GGT GTA GAA
*Pi-33*	8	Rm 72	F	CCG GCG ATA AAA CAA TGA G
			R	GCA TCG GTC CTA ACT AAG GG
		Rm310	F	CCA AAA CAT TTA AAA TAT CATG
			R	GCT TGT TGG TCA TTA CCA TTC
*Sub1A*	9	Sub1A203	F	GAT GT GT GGAGGAGAAGT GA
			R	GGTAGAT GCCGAGAAGT GTA
		Rm 7481	F	CGACCCAATATCTTTCTGCC
			R	ATTGGTCGTGCTCAACAAG
*SalTol*	1	Rm 493	F	GTACGTAAACGCGGAAGGTGACG
			R	CGACGTACGAGATGCCGATCC
		Rm 7075	F	TATGGACTGGAGCAAACCTC
			R	GGCACAGCACCAATGTCTC

**Table 2 plants-09-01580-t002:** Variations in the quantitative traits in F_2_ hybrids from crossing submergence-resistant samples with the variety Novator, 2014.

Trait	Crossing Combination			
	BR 11 × Novator	CR-1009 × Novator	Inbara 3 × Novator	TDK-1 × Novator
Plant height, cm	71–129 (97.5)	57–131 (89.4)	60–149 (100.2)	45–138 (99.9)
Panicle length, cm	11.5–27 (18.4)	10–26 (17.7)	9.5–32 (19.1)	9–27 (18.9)
Number of grains, pcs	10–220 (77.1)	2–201 (50.3)	4–343 (60.2)	4–180 (55.3)
Number of spikelets, pcs	57–322 (174.6)	38–273 (133.8)	18–411 (137.0)	17–261 (122.2)
Spikelet length, mm	6.1–10.1 (8.1)	6.8–9.8 (8.0)	6.1–11.9 (9.1)	7.2–11.3 (9.3)
Spikelet width, mm	2.3–3.8 (3.1)	2.5–4.0 (3.1)	2.1–3.8 (2.9)	2.3–3.9 (3.0)
Mass of 1000 grains, g	11–38 (25.4)	10–35 (23.2)	12–37 (25.9)	13–39 (25.8)
Mass of grain from the panicle, g	0.72–5.54 (1.98)	0.03–5.90 (1.22)	0.06–5.42 (1.54)	0.07–6.09 (1.79)

Note: The average value is indicated in brackets.

**Table 3 plants-09-01580-t003:** Selected F_2_ hybrid plants from crossing submergence-resistant samples with Novator, 2014.

Hybrid	Duration, Days	Plant Height, cm	Panicle Length, cm	Number of Grains in Panicle, pcs	Mass of 1000 Grains, g
Novator, st	112	97.5	16.5	110	31.8
176(BxN) *	120	108.0	21.5	146	27.0
334 (BxN)	118	96.7	17.3	145	21.6
34 (CxN)	121	81.2	16.5	109	25.3
390 (CxN)	122	82.0	17.5	122	27.7
273 (IxN)	120	95.4	15.1	151	24.4
507 (IxN)	119	97.2	19.0	138	29.3
81 (TxN)	123	96.5	17.2	159	32.0
393 (TxN)	121	97.3	14.9	152	25.2

Note *: (BxN), BR 11 × Novator; (CxN), CR-1009 × Novator; (IxN, Inbara 3 × Novator; (TxN), TDK-1 × Novator.

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
