# Peer review of "Rice Breeding in Russia Using Genetic Markers"

_plants, 2020, doi:10.3390/plants9111580_

Round 1

Reviewer 1 Report

  1. There is no need for Fig. 1,2, and 5 in the manuscript. They can be added as a supplementary file.
  2. Gel pictures are not crisp, I will suggest authors to add high-resolution figures.
  3. Add ladder size scale in the gel pictures.
  4. I will suggest authors to use the word paddy rather than rice.
  5. Why authors have not run the DNA ladder in Fig.3 and 4.
  6. Please see attached PDF for more corrections. 

Author Response

ЧЧЧЧСМПо

Пожалуйста, смотрите вложение в коробке

Reviewer 2 Report

I am attaching my comments as MS Word file.

Important note: Fig 3 and Fig 4 are identical gel photos!

Author Response

Пожалуйста, смотрите приложения в коробке

Reviewer 3 Report

The authors have studied the tolerance of Russian and Asian rice varieties to soil salinization, water flooding and blast, as well as studied the hybrids of the second and third generations from their crossing in order to obtain sustainable rice based on domestic varieties using DNA markers. DNA marker analysis by authors revealed 5 target genes in disease-resistant rice samples in a homozygous state. This is an interesting study important for the production of varieties epiphytotic development of the disease, preserving the biological productivity of rice and obtaining environmentally friendly agricultural products. However, I would like the authors to address the following minor queries to improve the manuscript.

  1. Language editing is highly recommended.
  2. As authors mentioned series of crosses, in which level of cross the pyramided blast resistance genes were found in a homozygous state? What was the segregation pattern here?
  3. If possible, please include the DNA ladder in figure 3and 4 to know the size of the DNA markers from the gel image?

Author Response

Пожалуйста, смотрите вложение в коробке
